# An integrated high-throughput robotic platform and active learning approach for accelerated discovery of optimal electrolyte formulations

Juran Noh[1,5], Hieu A. Doan [2,5] ✉, Heather Job[1], Lily A. Robertson[3], Lu Zhang[3], Rajeev S. Assary [2], Karl Mueller [4], Vijayakumar Murugesan [4] ✉ & Yangang Liang [1] ✉

Solubility of redox-active molecules is an important determining factor of the energy density in redox flow batteries. However, the advancement of electrolyte materials discovery has been constrained by the absence of extensive experimental solubility datasets, which are crucial for leveraging data-driven methodologies. In this study, we design and investigate a highly automated workflow that synergizes a high-throughput experimentation platform with a state-of-the-art active learning algorithm to significantly enhance the solubility of redox-active molecules in organic solvents. Our platform identifies multiple solvents that achieve a remarkable solubility threshold exceeding 6.20 M for the archetype redox-active molecule, 2,1,3-benzothiadiazole, from a comprehensive library of more than 2000 potential solvents. Significantly, our integrated strategy necessitates solubility assessments for fewer than 10% of these candidates, underscoring the efficiency of our approach. Our results also show that binary solvent mixtures, particularly those incorporating 1,4-dioxane, are instrumental in boosting the solubility of 2,1,3-benzothiadiazole. Beyond designing an efficient workflow for developing high-performance redox flow batteries, our machine learning-guided high-throughput robotic platform presents a robust and general approach for expedited discovery of functional materials.

The ability to design materials with targeted functional properties is critical for developing clean energy technology applications and to achieve deep decarbonization of electricity[1,2]. However, the conventional trial-and-error methods are costly and time consuming, and realizing new materials-based technologies typically requires 10–20 years of fundamental and applied research[3,4]. While data-driven methods based on machine learning (ML) have shown the potential to significantly accelerate the design of new materials for clean-energy technologies[5–9], their practical applications in materials research are still limited due to the scarcity of large and high-fidelity experimental databases[7,10].

Redox flow batteries (RFBs) have been shown as a leading technology to address the intermittent nature of renewable energy sources

[1]Energy and Environment Directorate, Pacific Northwest National Laboratory, Richland, WA 99354, USA. [2]Materials Science Division, Argonne National Laboratory, Lemont, IL 60439, USA. [3]Chemical Sciences and Engineering Division, Argonne National Laboratory, Lemont, IL 60439, USA. [4]Physical and Computational Sciences Directorate, Pacific Northwest National Laboratory, Richland, WA 99354, USA. [5]These authors contributed equally: Juran Noh, Hieu A. Doan. ✉e-mail: hadoan@anl.gov; vijay@pnnl.gov; yangang.liang@pnnl.gov

**Fig. 1 | Schematic of the closed-loop electrolyte screening process based on machine learning (ML)-guided high-throughput experimentation platform.** The workflow consists of a high-throughput experimentation module (top) connected with a Bayesian optimization module (bottom). The HTE components include automated sample preparation (top left) and characterization (top right), from which electrolyte properties such as solubility are obtained. The BO module utilizes experimental data to train a surrogate model (bottom right) and score potential candidates from a database using an acquisition function (bottom left). The top-ranking electrolyte formulations are suggested for the next round of experiments.

used for grid-scale energy storage[11]. Their unique design, which separates energy storage and power generation components, positions them competitively for long-duration storage needs[12–16]. Low cost redox-active organic molecules (ROMs) comprised of earth-abundant elements (C, N, H, O, S) are gaining attention as potential alternatives to their inorganic counterparts in RFBs[17]. However, a significant challenge for these systems lies in their reduced volumetric capacity, attributed to the low solubility of ROMs[18]. Hence, it is crucial to improve the solubility of ROMs to achieve a higher energy density in RFBs. In comparison to aqueous RFBs, nonaqueous RFBs (NRFBs) offer distinct advantages, including a wide operating temperature range, higher cell voltage, and the potential for increased energy density by tuning the solubility of ROMs in various organic solvents[19,20]. Nonetheless, developing highly soluble ROMs for NRFBs has proven to be a daunting task due to the lack of standardized and application-relevant experimental solubility data for organic solvent systems[21]. The ability to accurately determine the solubility of a solute in its saturated solution at equilibrium remains challenging as it depends on various factors including solute properties, solvent composition, equilibrium time, and temperature[21–23]. Such limitation impedes the success of data-driven design of electrolytes and subsequently NRFB research[12,24].

In general, solubility measurement is performed via 'excess solvent' or 'excess solute' methods[25]. The 'excess solvent' method involves gradual addition of the solvent to the solid until only a single liquid phase is observed. This allows for a quick determination of molar concentrations and enables the development of automated solubility screening systems using computer vision[25–27]. However, the 'excess solvent' method is a kinetic solubility measurement and while it is fast, its reliability is not always sufficient for high-fidelity data collection efforts. On the other hand, in the 'excess solute' method, saturated solutions are prepared and allowed to reach equilibrium prior to sample analysis. The 'excess solute' method is also known as the classical shake-flask method for thermodynamic solubility measurement. While this approach offers accurate and reproducible solubility measurement, the need for long incubation time and ex-situ analysis tools (HPLC, UV-Vis, and NMR) presents a critical hurdle for extensive data generation[25].

By leveraging an automated high-throughput experimentation (HTE) platform, it is possible to improve the reliability and efficiency of the 'excess solute' method and construct a solubility data library for NRFBs. This automated HTE approach has been envisioned to

simultaneously handle multiple samples, reducing incubation time per sample and minimizing chemical waste[28]. While generating high-quality solubility databases for molecules in organic solvents has become accessible thanks to recent advancements in robotics, it is still a time-consuming and laborious task for a couple of reasons[23,29]. First, the majority of existing HTE-based solubility determination methods were developed for aqueous systems[23,28,30]. Transitioning these methods to non-aqueous systems is not a straightforward task due to several hurdles, including chemical compatibility and volatility of organic solvents. Second, organic solvents can be utilized either in their pure form or as mixtures, offering nearly unlimited combinations. Indeed, solvent mixtures (e.g., binary solvents) are frequently used to enhance solubility and modify other properties through a synergistic effect[31–33]. In such cases, the solute demonstrates higher solubility in a binary solvent compared to pure solvents[31–33]. However, the large diversity of potential solvent mixtures also renders the screening process more time-consuming and expensive, even with HTE systems[33,34]. A strategic approach would be to develop an ML-guided HTE system for targeted and efficient solubility data generation for ROMs in organic solvent systems. Active learning (AL), particularly Bayesian optimization (BO), has been shown to be a reliable approach to accelerate the search for the desired electrolytes for energy storage applications[35]. Therefore, closed-loop experimental workflows guided by BO could be used to minimize HTE execution[36–39].

In this work, we use 2,1,3-benzothiadiazole (BTZ), a high-performance anolyte with highly delocalized charge density and good chemical stability[40,41], as a model ROM. The focus is on investigating its solubility in various organic solvents, demonstrating the potential of an ML-guided HTE robotic platform to accelerate the discovery of electrolytes for NRFBs. Specifically, we designed a closed-loop solvent screening workflow that consists of two connected modules, namely HTE and BO (Fig. 1). The HTE module carries out sample preparation and solubility measurement via a high-throughput robotic platform (see Experimental Methods). The BO component consists of a surrogate model and an acquisition function, both of which together serve as an oracle that makes solubility predictions and suggests new solvents for evaluation (see Computational Methods). Our workflow, as depicted in Fig. 1, is detailed in the following sequence of steps: Initially, we prepare saturated solutions and analytical samples of ROMs through the HTE platform. Next, we acquire nuclear magnetic resonance (NMR) spectra of these samples and

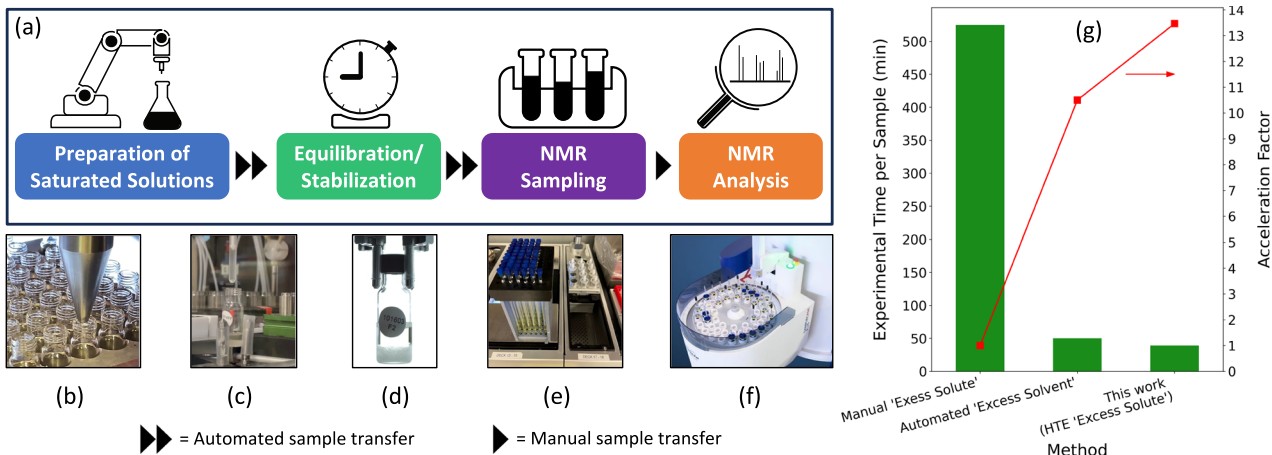

**Fig. 2 | Overview of the automated high-throughput experimentation (HTE) platform. a** Schematic representation of the automated HTE system for solubility measurement. The automation process consists of powder (**b**) and solvent (**c**) dispensing, (**d**) saturated sample monitoring, and nuclear magnetic resonance (NMR) sampling (**e**) and analysis (**f**). **g** Evaluated experimental time per sample for different solubility measurement methods. The data for automated 'excess solvent' method was estimated from the work of Shiri and co-workers[27]. The details on experimental time per sample are given in the SI (Supplementary Table S1).

employ the spectral data to calculate ROMs' solubility. This dataset is then used to train a surrogate model, which serves to predict the solubility of untested samples within our search space, as part of the BO process. Subsequently, we apply an acquisition function within the BO framework to guide the selection of new samples, directing our evaluation based on the balance of predicted solubility values and associated uncertainties, i.e., fitness score, thereby streamlining the discovery and analysis of potential solvents.

## Results and discussion

To generate high-fidelity and large-quantity solubility data for BTZ in organic solvents, we employed a highly automated, high-throughput sample preparation and characterization workflow (Fig. 2a). Our process starts with sample preparation wherein a robotic arm is used for powder and liquid dispensing (Fig. 2b, c, Supplementary Fig. S1, and Supplementary Video S1). Then, saturated solutions are allowed to stabilize at a fixed temperature for 8 hours to ensure thermodynamic equilibrium (Fig. 2d, Supplementary S2, and Supplementary Video S2). Following the stabilization period, liquids are automatically sampled into NMR tubes (Fig. 2e). Quantitative-NMR (qNMR) analysis is then carried out to determine molar solubility (mol L$^{-1}$) (Fig. 2f, see Methods for more details). Among those steps, the only manual operation is the transfer of NMR samples between the robotic platform and the NMR instrument. Overall, the automated platforms could prepare solute-excess saturated solutions and qNMR samples with minimal human intervention.

With our automated HTE workflow, the total experimental time to finish the solubility measurement for 42 samples is ca. 27 h (~39 min/sample, less time per sample with running more samples). As shown in Fig. 2g, this is more than 13 times faster than processing samples one by one manually using the 'excess solute' approach, which requires approximately 525 min per sample (Supplementary Table S1). While the screening speed of our HTE workflow based on the 'excess solute' method is comparable to that of the automated platform proposed by Shiri et al. (20–80 min/sample)[27], there are two important distinctions. First, we measured thermodynamic solubility, whereas Shiri and co-workers used the 'excess solvent' method for kinetic solubility measurements. Second, our workflow processes 42 or more samples at once, while Shiri et al.'s platform operates on one sample at a time.

In addition to the speed enhancement provided by the HTE system, we placed a strong emphasis on controlling experimental conditions, e.g., temperature (20 °C) and stabilization time (8 h), to ensure accurate measurements of BTZ solubility in various organic solvents.

Nevertheless, as shown in Supplementary Fig. S3a, we found slightly lower solubility values for BTZ in certain solvents compared to existing data[40]. This difference is likely attributed to variations in our methodology and the specific conditions of our experiments. Indeed, the influence of temperature on solubility highlights the need for standardized measurement techniques and comprehensive documentation of testing conditions. To ensure reproducibility, we also employed two control samples (2 M and saturated BTZ solutions in ACN) in every batch, particularly when repeat testing was not possible. The consistency of solubility values in these control samples across multiple batches, with a relative standard deviation of less than 5%, as shown in Supplementary Fig. S3b, validates the reliability and precision of our HTE approach, ensuring the generation of repeatable and high-fidelity data.

Based on our literature review and consideration of solvent properties[19], we made a list of 22 potential solvent candidates for BTZ (Table 1). Then, we further enumerated an additional 2079 binary solvents by combining those 22 single solvents in pairs, each with 9 different volume fractions (e.g., 0.1:0.9, 0.2:0.8, ...., 0.9:0.1). From this point we adopt a naming convention for our binary solvent systems such that S1:S2 @ f1:f2 denotes a mixture of solvent S1 and S2 at a volume fraction of f1 and f2, respectively (f1 + f2 = 1). As the surrogate model, i.e., Gaussian Process Regression (GPR), plays an important role in determining the performance of any BO approach, we first set out to evaluate the feasibility of using GPR for predicting the solubility of BTZ in various single and binary solvent systems. To create the training dataset, we carried out solubility measurement for all 22 single solvents and 36 randomly selected binary solvents of equal volume (listed in Supplementary Tables S2 and S3). Since each solvent sample consists of BTZ and up to two solvents, we considered a total of 11 relevant features derived from physicochemical properties and electronic structure calculations (DFT) of both the solvent and solute (e.g., molecular weight and topological polar surface area of a solvent molecule, computed maximum and minimum partial charge of a solvated BTZ molecule) as features for the GPR model (see Supplementary Table S4 for the complete list of features). The selection of such features was inspired by previous works[42,43] and further assessed by human experts.

The parity plot comparing GPR-predicted molarities to experimental measurements for our training set is shown in Fig. 3a. We observed a reasonable prediction accuracy with $R^2 = 0.81$, RMSE = 0.48 M, and MAE = 0.29 M. To test the generalizability of our GPR model, we picked and evaluated an additional set of 40 binary solvents

**Table 1 | List of 22 organic solvent candidates and their physicochemical properties**

| Full name | Abbr. | Formula | B.P.[a](°C) | M.P.[b](°C) | $\rho$[c](g mL$^{-1}$) | $\varepsilon$[d] |
|---|---|---|---|---|---|---|
| Dimethoxyethane | DME | $C_4H_{10}O_2$ | 84.5 | −58 | 0.868 | 7.2 |
| Dioxane | DOX | $C_4H_8O_2$ | 101.3 | 11.8 | 1.03 | 2.25 |
| Acetonitrile | CAN | $C_2H_5N$ | 81.6 | −45 | 0.786 | 37.5 |
| *N,N*-Dimethylformide | DMF | $C_3H_7NO$ | 153 | −61 | 0.944 | 36.71 |
| *N,N*-Dimethylacetamide | DMA | $C_4H_9NO$ | 166.1 | −20 | 0.94 | 37.78 |
| Propylene carbonate | PC | $C_4H_6O_3$ | 240 | −48.8 | 1.2 | 64 |
| *n*-Butylacetate | BA | $C_6H_{12}O_2$ | 126.1 | −78 | 0.882 | 5.01 |
| Dimethyl sulfoxide | DMSO | $C_2H_6OS$ | 189 | 19 | 1.1 | 46.68 |
| Cyclohexane | CH | $C_6H_{12}$ | 80.7 | 6.47 | 0.779 | 2.02 |
| Toluene | TOL | $C_7H_8$ | 110.6 | −95 | 0.867 | 2.38 |
| Heptane | HP | $C_7H_{16}$ | 98 | −90.55 | 0.684 | 1.92 |
| Octane | OT | $C_8H_{18}$ | 125 | −57 | 0.703 | 1.95 |
| 1-Propanol | PA | $C_3H_8O$ | 97 | −126 | 0.803 | 21 |
| *p*-Xylene | PX | $C_8H_{10}$ | 138 | 13.2 | 0.861 | 2.35 |
| *m*-Xylene | MX | $C_8H_{10}$ | 138 | −48 | 0.86 | 2.37 |
| *o*-Xylene | OX | $C_8H_{10}$ | 143 | −24 | 0.88 | 2.57 |
| *N*-Methyl pyrrolidinone | NMP | $C_5H_9NO$ | 202 | −24 | 1.03 | 32.3 |
| Cyclohexanone | CHO | $C_6H_{12}$ | 154 | −31 | 0.995 | 16.1 |
| Hexamethylphosphoramide | HMPA | $C_6H_{18}N_3OP$ | 157 | 7.2 | 0.779 | 30.54 |
| Butyronitrile | BTN | $C_4H_7N$ | 230 | −111.9 | 1.03 | 24.83 |
| Adiponitrile | APN | $C_6H_8N_2$ | 115 | 1 | 0.794 | 32.73 |
| Glutaronitrile | GTN | $C_5H_6N_2$ | 295 | −29.6 | 0.951 | 35.1 |

[a]Boiling point.
[b]Melting point.
[c]Volumetric density.
[d]Dielectric constant.

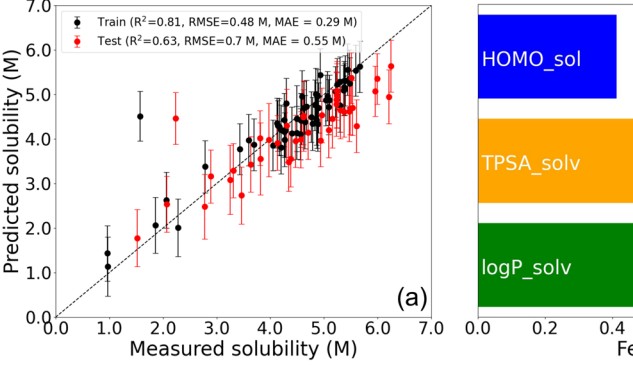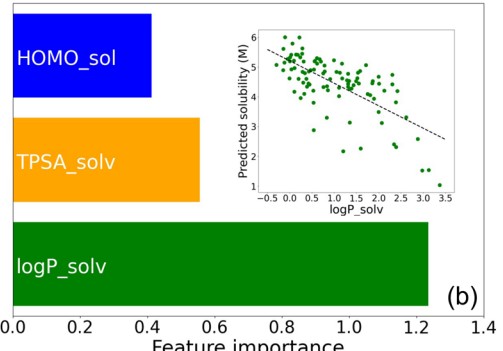

**Fig. 3 | Gaussian Process Regression (GPR). a** Parity plot of GPR-predicted solubility versus measured values from experiment. The error bars represent one standard deviation. **b** List of the most important features obtained from feature permutation analysis of the GPR model. HOMO_sol, TPSA_solv, and logP_solv represent the computed energy of the highest occupied molecular orbital of the solute molecule, the calculated topological polar surface area of the solvent molecule, and octanol-water partition coefficient value of the solvent, respectively. The inset shows the correlation between GPR predictions and the most important feature (logP_solv). The black dashed line indicates the linear fit of predicted solubility with respect to logP_solv. Source data are provided as a Source Data file.

(Supplementary Table S5). This test set was selected via Latin hypercube sampling to maximize its diversity[36]. As expected, the model is less accurate on the test set ($R^2 = 0.63$, RMSE = 0.7 M, MAE = 0.55 M) as compared to the training set. Regardless, given the fact that our GPR model was trained on only ca. 3% of the entire search space (58 out of 2101 solvents), we found such performance satisfactory. In addition, the octanol-water partition coefficient value of the solvent (logP_solv) is identified as the most important feature of the GPR model based on feature permutation analysis, and a correlation between GPR-predicted solubility and logP_solv is indeed observed (Fig. 3b, inset). Here, logP_solv represents the octanol-water coefficient for a solvent, serving as a means to assess the polarity disparity between that solvent and water. If we consider the polarity of water as our baseline, given that the polarity variance between BTZ and water remains constant, then logP_solv effectively characterizes the polarity distinction between a solvent and BTZ. Similarly, TPSA_solv, denoting the topological polar surface area of a solvent molecule, indirectly offers insights into the localized polarity contrast between a solvent and BTZ. These findings are consistent with the current knowledge of solubility as a function of polarity, as in the general solubility equation proposed by Yalkowsky et al.[42]. These findings are consistent with the current knowledge of solubility as a function of polarity, as in the general solubility equation proposed by Yalkowsky et al.[42].

The observed performance of the GPR model provides confidence in its usage as the surrogate model in a BO workflow for identifying solvents with the desired solubility of BTZ. Before deploying a BO

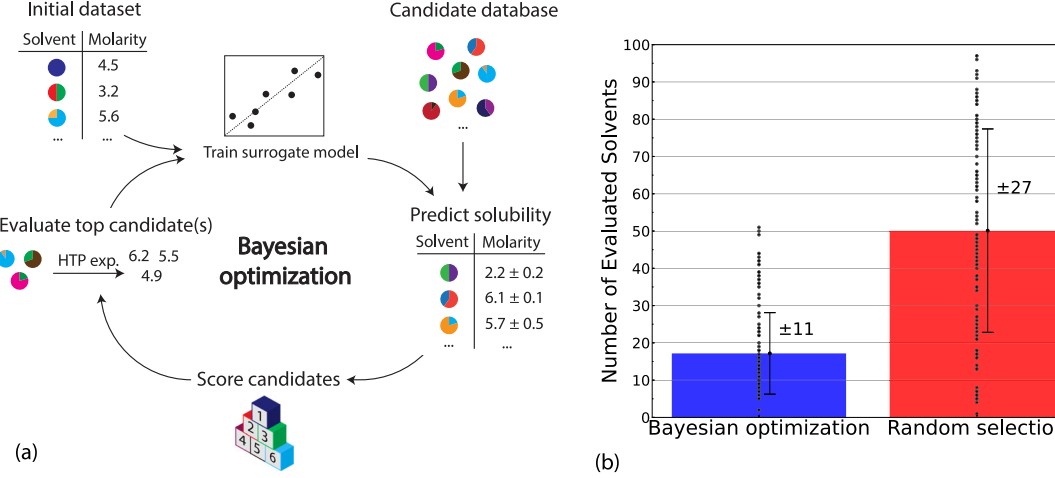

**Fig. 4 | Identification of desired electrolytes via Bayesian optimization (BO). a** Schematic representation of Bayesian optimization algorithm for accelerated screening of binary solvents with desired solubility of BTZ. The algorithm starts with training a surrogate model using an initial dataset of measured solubilities. The resulting model is then employed to predict solubilities of solvent candidates from a database. Based on these predictions and their uncertainties, unknown candidates are ranked according to their potentials to improve over their known counterparts. The top-ranking solvents are suggested for experimental evaluations, from which new data are obtained and used for re-training a new surrogate model in the next round of BO. **b** Comparison between BO and random selection for the number of required evaluated solvents before identifying the one with highest solubility of BTZ from the 98 solvent dataset. The height of the color bar and black error bar represent the mean and standard deviation of 100 trials. Source data are provided as a Source Data file.

model for the remaining candidate library of 2003 binary solvents, we first performed a benchmarking experiment using the current known dataset of 98 solvents. The goal of this experiment was to evaluate the effectiveness of BO in identifying the solvent with the highest solubility of BTZ, namely DOX:DMF @ 0.6:0.4, out of all 98 solvents in the dataset. A schematic representation of our BO algorithm is shown in Fig. 4a. Initially, a set of 5 randomly selected solvents were evaluated and their corresponding solubilities of BTZ were used to train a GPR surrogate model for solubility prediction. This model was then employed to predict the solubility of BTZ, with quantified uncertainty, for the remaining 93 solvents. Based on the predicted values, an acquisition function, namely expected improvement (EI), was used to rank 93 solvents for their potential to maximize solubility. The solvent with the highest EI-score is then evaluated, and subsequently added to the training set. This completes the first loop of BO, and the second loop begins with six training datapoints and 92 remaining candidates. The iterative process is continued until DOX:DMF @ 0.6:0.4, the solvent with the highest BTZ molarity of 6.25 M, is found. To generate reasonable statistics for the performance of BO, we repeated our experiment 100 times with different initial sets of five randomly selected solvents. Our results indicate that on average BO identifies DOX:DMF @ 0.6:04 after suggesting a total of 17 ± 11 out of 98 solvents for solubility evaluation (Fig. 4b). For comparison, random selection requires approximately 50 ± 27 solubility measurements to find the same solvent. We also performed t-test on the two distributions and obtained a $p$-value of $1.17 \times 10^{-20}$, indicating that the performance improvement of BO over random selection is statistically significant. Furthermore, different acquisition functions such as Thompson sampling, upper confidence bound, and probability of improvement can also be used with BO to identify the optimal solvent more quickly than random selection; however, EI is shown to have the best performance among them (Supplementary Fig. S4). Overall, we found BO to be a robust and efficient approach for accelerating the identification of solvent candidates with the desired solubility of BTZ.

For the final screening that aims at identifying the binary solvent systems with highest solubility of BTZ among the remaining 2003 candidates, all 98 labeled samples were used to initialize the BO model. By employing a similar workflow as shown in Fig. 4a, we carried out a total of three BO cycles, wherein 40 solvent samples were suggested and evaluated per cycle (Supplementary Tables S6–S8). As shown in Fig. 5a, after the first cycle (1$^{st}$ batch), we discovered a new binary composition, namely DOX:DMSO @ 0.8:0.2, with a higher solubility of BTZ than that of the best solvent (DOX:DMF @ 0.6:0.4) in the initial set of 98 solvents (6.50 M vs 6.25 M). In addition, the solubility distribution is more concentrated at higher values for solvents in the first batch compared to those in the initialization set, and the trend continues as more cycles were carried out. However, the median and maximum molarities that peak in the first cycle slightly decrease in the subsequent cycles (from 5.98 M to 5.88 M to 5.69 M for median molarity and from 6.50 M to 6.45 M to 6.25 M for maximum molarity). We hypothesize that, since the best binary solvent composition which is DOX:DMSO @ 0.8:0.2 had already been identified in first cycle, only solvent candidates with lower solubility (<6.50 M) were found in subsequent cycles. More importantly, we were able to identify 18 new binary solvent systems with solubilities of BTZ greater than 6.20 M, after conducting only 218 measurements from over 2,000 potential candidates. The solubility values of the top five binary solvents, depicted in Fig. 5b, are quite similar, ranging from 6.40 to 6.50 M. It is also noted that this list is biased towards DOX-containing mixtures, which is reasonable as DOX possesses the highest solubility of BTZ (5.47 M) among all single solvents. We believe that the value of our BO model lies in its ability to exploit the synergistic effects in solvent mixing that cannot be easily perceived by chemical intuition. As it is shown in Fig. 5b, all binary solvents yield markedly higher solubility for BTZ compared to that of their constituents. Notably, the combination of DOX with GTN, a low solvating solvent (1.86 M), leads to an unexpectedly and highly soluble system for BTZ at 6.48 M. While the current model is robust for solubility prediction of BTZ in binary solvents, we recognize it is necessary to further extend its application toward more complex systems of more than two components, as practical NRFB electrolytes also include supporting salts and other organic species. In addition, since solubility is not the only property that affects the electrochemical performance of electrolytes in NRFBs, future generations of the ML-guided HTE platform should account for other important factors such as viscosity, ionic conductivity, and chemical stability.

In summary, we have showcased an ML-guided HTE platform for electrolyte screening wherein ML predictions and automated experiments work in unison to efficiently screen for binary organic solvents

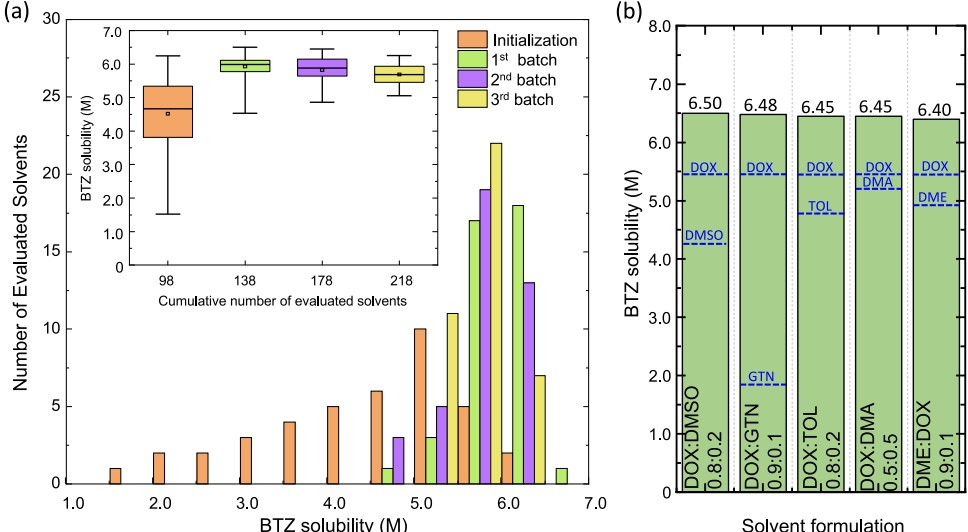

**Fig. 5 | Results of the closed-loop solvent screening workflow. a** Distribution of measured solubilities of 2,1,3-benzothiadiazole (BTZ) among all investigated solvents in this work. In the inset, the bottom and top whiskers represent the minimum and maximum values, respectively. The length of the box indicates the interquartile range from the first (lower side) to the third (upper side) quartile. The median and mean are the horizonal line and square inside the box. **b** Five binary solvent formulations with the highest solubilities of BTZ (green bar). Solubility values of their respective pure components are shown as blue dashed lines. Source data are provided as a Source Data file.

with optimal solubility for BTZ. With this platform, we successfully identified 18 binary solvent systems with BTZ solubility surpassing 6.20 M after conducting measurements for only 218 out of 2101 candidates. In the process, we constructed a highly standardized solubility database encompassing diverse organic solvents, allowing for further development of ML methods for solubility prediction. Our work not only serves to connect the fields of data science and traditional experimental science but also lays the groundwork for the future development of an autonomous platform dedicated to battery electrolyte screening.

# Methods
## Materials
2,1,3-benzothiadiazole (>99.0%) and 1,4-dinitrobenzene (>99.0%) were purchased from TCI America. *p*-xylene, *m*-xylene, *o*-xylene, hexamethylphosphoramide and butyronitrile were purchased from Sigma Aldrich. Cyclohexanone was purchased from TCI America. Other solvents (>99.0% with extra dried condition) were purchased from Acros Organics and used without any pretreatment.

## Preparation of saturated solutions using a high-throughput automated platform
The saturated solutions were prepared by a robotic platform (Big Kahuna, Unchained Labs) as shown in Supplementary Fig. S1. Experiment designs were programmed using the software, Library Studio Ver 9.2 (Unchained Lab). BTZ (TCI America, >99.0%) was first dispensed into 2 ml vials following with prime solvent and secondary solvent as shown in the experimental design (Supplementary Fig. S5). Sample solutions in 40 various formulations and two control sample solutions (2.0 M and saturated of BTZ in ACN) (42 solutions total) were prepared in one 48-vial microplate (Supplementary Fig. S5, top).

The whole powder and liquid dispensing process was performed in the argon filled glove box. Immediately after solvent dispensing, the vials were capped to prevent undesired evaporation. The capped vials were vortexed at 1,000 RPM and stirred at 500 RPM for 1–3 h to prepare the 'excess solute' solutions, with confirmation of any undissolved solid solute achieved by an on-line vision system (Supplementary Fig. S6a). Subsequently, the vials were placed on the deck setting at 20 °C for 8 h to reach equilibrium (Supplementary Table S9). Once the BTZ

solutions reached equilibrium, some BTZ crystals precipitated at the bottom (Supplementary Fig. S2), and the supernatant (top clear solution) was used for qNMR analysis.

## Solubility measurement via quantitative H-NMR spectroscopy
Quantitative $^1$H NMR spectroscopy, utilizing 1,4-dinitrobenzene (DNB) as an internal standard (referred to as INSD), was employed to measure the concentration. The NMR sampling process was done automatically on the robotic platform. Firstly, DNB was dissolved in deuterated dimethyl sulfoxide (DMSO-$d_6$, Acros Organics) to prepare an 8.00 mg mL$^{-1}$ INSD bulk solution and placed on the source deck. The capped sample vials were uncapped while transferring 30 μL of each saturated solution from liquid phase to NMR tubes (Wilmad-Labglass, USA) (Supplementary Fig. S6c, d). During the transfer process, aspiration was slowly conducted to avoid undesired suction of BTZ solid precipitates. After transferring the samples, 600 μL of the INSD solution was dispensed into each NMR tube, and the tubes were capped. Before $^1$H NMR measurement, the NMR tubes were shaken thoroughly to ensure the homogeneous mixing. The $^1$H NMR spectra were obtained using a Bruker 400 MHz Avance III NMR equipped with SampleCase (Autosampler). The molar concentrations of BTZ were calculated by comparing the integrated area ratio with the INSD using Eq. (1):

$$C_{BTZ,sat\,solution} = \frac{V_{NMR\,sample}}{V_{sat.sol.}} \bullet \frac{I_{BTZ}}{I_{INSD}} \bullet \frac{N_{INSD}}{N_{BTZ}} \bullet C_{INSD,NMR} \quad (1)$$

where $N_{BTZ}$ (= 4) and $N_{INSD}$ (= 4) are the number of hydrogen atoms in BTZ and DNB (INSD), respectively. As shown in Supplementary Fig. S7, the hydrogen atoms in DNB are labeled as 'a', whereas those in BTZ are distinguished as 'b' and 'c'. Subsequently, $I_{INSD}$ is the integrated area of peak 'a' and $I_{BTZ}$ is summation of integrated areas of peak 'b' and 'c'. $C_{INSD,NMR}$ is the molar concentration of INSD in the NMR solution, which consists of 30 μL of BTZ sample solution ($V_{sat.sol.}$) and 600 μL of INSD bulk solution, totaling 630 μL ($V_{NMR\,sample}$). Prior to the production runs, we prepared two reference samples in acetonitrile (ACN) with target BTZ concentrations of 1.0 M and 2.0 M to evaluate the accuracy of our automated workflow and qNMR analytical method. The solubilities of BTZ calculated from NMR spectra were found to be 0.98 M and 1.98 M (Supplementary Fig. S7), indicating the accuracy of our approach.

## High-throughput viscosity measurement

We developed a high-throughput viscosity measurement workflow by integrating automated sampling on our robotic platform (100 μL saturated solution into a 2 ml vial) with a high-throughput viscometer (VROC® initium one plus, RheoSense) (Supplementary Fig. S8b). According to our analysis, viscosity displays minimal sensitivity to the concentration of BTZ, resulting in an increase of less than 2 cP in solutions. Notably, the majority of saturated solutions exhibits viscosity values below 2.5 cP, as illustrated in Supplementary Fig. S8a.

## Machine learning

**Feature generation.** To create an accurate model to predict solubilities of BTZ for unary and binary solvents, we employed several relevant physicochemical descriptors including molecular weight, topological polar surface area, number of heavy (non-hydrogen) atoms, and octanol-water partition coefficients of the solvent molecules (logP_solv). In addition, we carried out first-principles simulations of solvated BTZ molecules in different solvents to compute solute-related descriptors such as solvation free energies, dipole moments, polarizability, HOMO and LUMO energies, maximum and minimum partial charges. A total of 11 features were tabulated in Supplementary Table S4. For simplicity, descriptor values of a binary solvent are calculated by combining those of its constituents weighted by their corresponding mol fractions.

**Gaussian process regression.** A Gaussian Process (GP) is a collection of random variables, any finite number of which have a joint Gaussian distribution[44]. A GP is completely specified by its mean function $m(x)$ and covariance function (or kernel) $k(x,x')$, and can be written as:

$$f(x) \sim GP(m(x), k(x,x')) \tag{2}$$

If x and x' represent the feature vectors, then their covariance based on the Matérn kernel ($v = 1.5$) is expressed as follows:

$$k(x,x') = \left(1 + \frac{\sqrt{3}|x-x'|}{\sigma_l}\right) * \exp\left(-\frac{\sqrt{3}|x-x'|}{\sigma_l}\right) + \sigma_n^2 \tag{3}$$

Here, $\sigma_l$ and $\sigma_n$ are the length scale and the expected noise level in the data set, respectively. Each parameter was determined using the maximum likelihood estimate during model training.

**Expected improvement (EI) acquisition function.** The EI acquisition function was given by the following equation[36,45]:

$$EI(x) = \begin{cases} (\mu(x) - f(x^+) - \varepsilon)\Phi(Z) + \sigma(x)\phi(Z) & \sigma(x) > 0 \\ 0 & \sigma(x) = 0 \end{cases} \tag{4}$$

$$Z = \frac{\mu(x) - f(x^+) - \varepsilon}{\sigma(x)} \tag{5}$$

where $\mu(x)$ and $\sigma(x)$ are the predicted mean and standard deviation from the GPR model, $f(x)$. $\Phi(Z)$ is the cumulative density function (CDF), and $\phi(Z)$ is the probability density function (PDF). $f(x^+)$ is the predicted property of the current best material, and $x^+$ is the feature vector of that material. In Eqs. (4) and (5), a constant $\varepsilon$ value of $10^{-2}$ was used to balance the trade-off between exploitation (pursuing the trend of the current best estimates) and exploration (diversifying the search to avoid local optima).

## Density Functional Theory (DFT)

All DFT simulations were performed using Gaussian 16 software[46] at the b3lyp/6-31 + G(d,p)[44] level of theory. Numerical integrations were carried out using the ultrafine grid. To compute the properties of BTZ in 22 unary solvents, self-consistent reaction-field (SCRF) calculations using the Polarizable Continuum Model (PCM) were employed (See Table 1 for the list of dielectric constants). The Gibbs free energies of BTZ (at 298 K) in the gas phase ($G_{BTZ,gas}$) and in the solvent ($G_{BTZ,solvent}$) were used to calculate the solvation free energy ($G_{solv}$) in each of the solvents via the following equation:

$$G_{solv} = G_{BTZ,solvent} - G_{BTZ,gas} \tag{6}$$

## Reporting summary

Further information on research design is available in the Nature Portfolio Reporting Summary linked to this article.

## Data availability

All data generated in this study are provided in the Supplementary Information Source Data file. Source data are provided with this paper.

## Code availability

Code is available on Zenodo[47] and Github repository: https://github.com/MolecularMaterials/AL-HTE-Electrolyte.

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

## Acknowledgements

The research was financially supported by the Joint Center for Energy Storage Research (JCESR), an Energy Innovation Hub funded by the U.S. Department of Energy, Office of Science, Basic Energy Sciences. We also acknowledge the support from the Automated Robotics for Energy Storage Laboratory (ARES Lab) funded by the Energy Storage Materials Initiative (ESMI), which is a Laboratory Directed Research and Development Project at Pacific Northwest National Laboratory (PNNL). The submitted manuscript has been created by Pacific Northwest National Laboratory and Argonne National Laboratory, which are U.S. Department of Energy Office of Science laboratories.

## Author contributions

Y.L., V.M., and H.A.D. conceived the research. J.N., H.A.D., H.J., and Y.L. designed and conducted the experiments. H.A.D developed the active learning/Bayesian optimization code and performed DFT calculations for quantum chemistry-derived machine learning features. J.N., H.A.D., and Y.L. wrote the manuscript. L.A.R., L.Z., R.S.A. and K.M. contributed to the manuscript revision and data analysis. All authors have given approval to the final version of the manuscript.

## Competing interests

The authors declare no competing interests.
