## [Peer Review File · Nature Communications]

An Integrated High-throughput Robotic Platform and Active Learning Approach for Accelerated Discovery of Optimal Electrolyte FormulationsREVIEWER COMMENTS

Reviewer #1 (Remarks to the Author):

In the frame of this paper, authors present a high-throughput experimentation platform complemented with active learning approach for optimization of the solubility of redox active molecules in organic electrolytes for NRFB application. Although scope of the Nature Communications journal, presented and discussed work is not on an adequate scientific level and there are too many open points related to the presented system, workflow and approach in general. The novelty of this work is low. The description of the HTE system is not complete. Level of "full" automation is not completely clear: It remains unclear which steps are manual, and which automated both in preparation and measurements? Why do authors consider 42 solutions? How many repetitions were made? How many failed? The main criticism relates to the lack of proof of concept in terms of identified lead electrolyte candidates and corresponding cell performance, which is a major shortcoming of this manuscript and cannot be accepted as completed evidence of AI-driven HTE selection of electrolytes for NRFBs. Electrochemical terms are not properly used throughout the manuscript. The number of references should be increased. The English needs to be revised. As a result, the paper in its present form is not at all suitable for publication in Nature Communications and should be rejected.

Reviewer #2 (Remarks to the Author):

The manuscript by Noh et al. presents high-throughput screening of solubilities of 2,1,3-benzothiadiazole as a model ROM for organic redox flow batteries by a robotic platform, in which the authors adopt Bayesian optimization to assist the evaluation and exploration of unary and binary mixtures. The study demonstrates that AI-guided robotic platform can efficiently look for non-intuitive combinations of solvent electrolytes for energy applications. The work is of significance to the battery community, but the following concerns need to be addressed.

1. The authors should provide the brand and model of the automated deck and clearly state your own contributions, i.e. how you (1) interfaced the vision system and the qNMR with the deck and (2) developed the GPR model for checking the solubility of BTZ. These details would help highlight contributions to the field.
2. In Fig 3, it would be nice to include error bars to the obtained values.
3. How was the 8-hour sample stabilization time decided? Did you have a calibration curve or choose the value randomly? Since this step takes the longest time among all steps, optimizing it could potentially shorten the overall time, hence the screening rate.
4. It is to be reminded that high solubility is not the only criterion for high energy redox flow batteries. Other physicochemical and electrochemical properties, such as the viscosity, stability, and conductivity, should be carefully evaluated after identifying the sample mixture.
5. By the way, the authors should correct all "physiochemical" to "physicochemical", as the two words have different meanings.

Reviewer #3 (Remarks to the Author):

Authors report on an integrated robotic ML guided setup to optimize solubility on a model redox additive. Solids handling is tricky in automated systems and authors have done a good job in closing the loop. A few comments below:

- 1) Figure 1 needs to be improved. The figure itself is not of the right quality and caption description does not tell me how the system is closed loop. Please make this a clear system flow diagram and provide detailed caption.
- 2) Figure 2f, recommend adding the acceleration factor as a second y-axis.
- 3) I'd recommend dropping AI term used in the paper and modify it to ML or Active Learning as described in the abstract. There is very narrow set of machine learning done here.
- 4) How do the authors plan to give the dataset for the measurements? I'd recommend having a clear schema in a csv file that can easily be ingested in standard programming frameworks. I'd recommend SMILES, SELFIES or CAS Number or some other unique representation for the molecules in the system.
- 5) Did you try other acquisition functions - Thompson sampling, upper confidence bound, in addition to EI?
- 6) A few other non-technical comments: I'm a little disappointed that Library Studio Ver 9.2 (Unchained Lab) has been used for experiment design rather than some open-source framework. I recognize that the system was purchased, so this decision had to be taken but this makes it very difficult to reproduce the results.
- 7) The title needs to be changed - please stress on the problem you are solving. The system is specifically focused on solubility - the title is overclaiming in terms of "electrolyte discovery". This system and other closed loop systems are only capable of optimization, discovery would require the ability to move through chemical space - which is beyond the capability of current ML-guided automated experimentation setups.

Overall, the authors have done an impressive job to push on solids/liquids handling and closing the loop related to electrolyte optimization.

Reviewer #4 (Remarks to the Author):

First, thank you to the authors for submitting their manuscript "An Integrated High-throughput Robotic Platform and Active Learning Approach to Accelerate Electrolytes Discovery" to the Nature Communications journal. The article presents an interesting integrated workflow combining high-throughput robotic experimentation and active learning to accelerate the discovery of organic electrolytes for redox flow batteries. The topic is relevant, and the approach is novel for accelerating the design of electrolytes for redox flow batteries.

The authors can find proposed corrections, comments, and amendments below:

** Abstract:

- "... Physiochemical property ..."  "... Physicochemical property ..."
- "... to generate and optimize..."  "... to optimise ...". The term "generate" is here misleading.

** Introduction:

- Shall we introduce the introduction with an "Introduction" section title? Naïve question regarding the expected manuscript's format in the journal.
- "... cycle life ..."  "... lifecycle ..."
- "... , known as ROMs, ..."  "... (ROMs) ..."

- "... high throughput ..."  "... high-throughput ..."
 - "... facile ..."  "... accessible ..."
 - "... The majority of high-throughput experimentation (HTE) ..."  Acronym already introduced earlier. Please use it. Overall, a third or fourth reading of the main text would have avoided many straightforward corrections.
 - "... mixtures for a given organic molecule is practically infinite ..."  "... mixtures for a given organic molecule is increased ..."
 - "... which increases the chances of finding a suitable combination but also makes the screening process more time-consuming and expensive even with HTE systems. ..." \diamond "... which makes the screening process more time-consuming and expensive even with HTE systems. ...". The claim on an increased probability of positive findings with expanding the space of mixture possibilities isn't supported by any data or principle here.
 - In step 4 of the step-by-step workflow description, please briefly add the role of the acquisition function here as you introduce the surrogate model in step 3.
 - "As the surrogate model...": "e.g." should be "i.e."
 - As a general remark on references. Why don't you refer to "S. Matsuda, G. Lambard, K. Sodeyama, "Data-driven automated robotic experiments accelerate the discovery of multi-component electrolyte for rechargeable Li-O₂ batteries", Cell Reports Physical Science, Volume 3, Issue 4, 2022, 100832, <https://doi.org/10.1016/j.xcrp.2022.100832>." for which the setup shares high similarities with the one detailed in the present manuscript.
 - The authors claim the methodology determines "thermodynamic" solubility, but the details suggest it is closer to kinetic solubility. Thermodynamic solubility requires proper equilibrium, which is difficult to confirm in high-throughput experiments with short timescales. This distinction should be clarified.
 - Solubility depends strongly on temperature, as the authors pointed out, but limited temperature data is provided. How was temperature controlled and measured? The impact of slight temperature variations should be considered.
- ** Results and discussion:
- Figure 2-f needs more consistency in uncertainty quantifications. Because of this, it's hard to assess the viability of the presented methodology compared to the two others, even though the absolute difference with a manual experiment is still expected to be significant. Please upgrade the figure accordingly as far as possible. If the authors struggle to assert the uncertainty of a given methodology, the reason should be pointed out in the main text.
 - "..., an the workflow ..."  "..., a workflow ..."
 - "(Fig. 2a) Powder and liquid dispensing systems (Fig. 2b and 2c) were integrated ..."  This is the main text. Please update the format you use to introduce the figures, e.g., "Powder and liquid dispensing systems (Fig. 2b and 2c) were integrated into the workflow (Fig. 2a) ..."
 - "... vision-tracking ..."  "... vision-tracking ..."
 - "... acetonitriltheACN) ..."?
 - "... the NMR spectra were comparable to the target concentration ..." \diamond Please be more specific in the comparison. Quantify it and share it in the main text.
 - "... conditionstheur ..."?

- Fig. 3-a, b: Is it possible to assess the uncertainty on the measured solubility per solvent (3-a) and per batch (3-b)?

- "... miscible. (Fig. S3). From ..."  "... miscible (Fig. S3). From ..."

- "... we considered a total of 11 relevant features derived from physiochemical properties and electronic structure calculations (DFT) of both the solvent and solute ..."  How did you choose the relevant features?

- "... the partition coefficient (logP) value of the solvent was identified as the most important feature of the GPR model ..."  Is this statistical correlation of the solubility with the LogP consistent with the actual physicochemical knowledge? And if yes, why? Can you interpret the result of this feature's importance distribution? As well as for 'TPSA' and the 'HOMO' level.

- Can you explain the origin of the error bars in Fig. 4-a? Possibly as an intrinsic consequence of the GPR model?

- "... Bayesian optimisation (BO) ..."  You don't need to re-introduce the acronym here as already done in the introduction.

- "... average BO identifies DOX:DMF @ 0.6:04 after suggesting a total of 17 out of 98 solvents for solubility evaluation ..."  What's the uncertainty on 17 (i.e. from Fig. 5-b) as you indicated it for the random selection (50+/-27)? Consistent uncertainties quantification is the root of statistically well-supported comparisons.

- The data analysis comparing random selection and Bayesian optimisation needs more rigour. The authors should perform statistical tests (e.g., compute p-value) to determine if the difference between both methods is significant. Error bars in Fig. 5-b are indeed substantial.

- "... as shown Fig. 5a ..."  "... as shown in Fig. 5a ..."

** Computational methods:

- Feature generation: "In addition, we carried out first-principles simulations of solvated BTZ molecules in different solvents to compute solute-related descriptors such as ..."  Can you assess the time needed to perform the DFT calculations for the initialisation set and per batch? The time should be added to the time needed for the automated protocol in the manuscript for the initialisation set and per batch. In this case, the computation time needed to apply the BO should be relatively insignificant.

- "... exploitation and exploration ..."  Please be a bit more specific on what those terms mean for non-specialists in the field of ML.

Overall, this paper shows promise and requires revisions to address the comments above. If these issues are discussed thoroughly, the article could become suitable for publication after another round of review.

Response Letter to Reviewers' Comments on Manuscript

NCOMMS-23-28903

We thank the reviewers for their valuable comments. In response, we conducted experiments to perform high-throughput viscosity measurements of the saturated solutions, diligently addressed each comment and incorporated the suggestions into the revised manuscript (*major changes highlighted in yellow for clarity*), as explained below in detail. Answers to the reviewers' comments are presented in blue type, while a portion of the updates to the manuscript and Supplemental Information (SI) is copied here and highlighted in yellow.

Reviewer 1

In the frame of this paper, authors present a high-throughput experimentation platform complemented with active learning approach for optimization of the solubility of redox active molecules in organic electrolytes for NRFB application.

Although scope of the Nature Communications journal, presented and discussed work is not on an adequate scientific level and there are too many open points related to the presented system, workflow and approach in general. The novelty of this work is low. The description of the HTE system is not complete. Level of "full" automation is not completely clear: It remains unclear which steps are manual, and which automated both in preparation and measurements? Why do authors consider 42 solutions? How many repetitions were made? How many failed? The main criticism relates to the lack of proof of concept in terms of identified lead electrolyte candidates and corresponding cell performance, which is a major shortcoming of this manuscript and cannot be accepted as completed evidence of AI-driven HTE selection of electrolytes for NRFBs. Electrochemical terms are not properly used throughout the manuscript. The number of references should be increased. The English needs to be revised.

As a result, the paper in its present form is not at all suitable for publication in Nature Communications and should be rejected.

Response:

We appreciate the reviewer's time and effort to assess our manuscript. We understand your concerns and have carefully considered your valuable feedback. We would like to provide some clarifications and address the points you have raised.

Regarding the scope and novelty of our work, we respectfully disagree with the assessment of its significance. Our research sheds light on the intricacies of measuring solubility, particularly within the context of redox flow battery applications. The integration of our high-throughput experimentation platform with an active learning approach serves as a significant step forward, providing a versatile tool for identifying energy materials with desired properties in a cost-effective and efficient manner. By addressing the challenges associated with solubilization processes influenced by various factors such as solute properties, solvent composition, equilibrium time, and temperature, our study contributes not only to the field of solubility measurement but also to the advancement of current understanding in the domain of redox active molecules in organic

electrolytes. We believe that our work has the potential to offer valuable insights into the development of standardized methodologies for measuring thermodynamic solubility and may have broader implications for future research in this area. (“Solubility Challenge Revisited after Ten Years, with Multilab Shake-Flask Data, Using Tight (SD ~ 0.17 log) and Loose (SD ~ 0.62 log) Test Sets” *J. Chem. Inf. Model.* 2019, 59, 6, 3036–3040 DOI: 10.1021/acs.jcim.9b00345; “Solubility challenge: Can you predict solubilities of 32 molecules using a database of 100 reliable measurements?” *J. Chem. Inf. Model.* 2008, 48, 1289– 1303, DOI: 10.1021/ci800058v; “Predicting Solubility Limits of Organic Solutes for a Wide Range of Solvents and Temperatures” *J. Am. Chem. Soc.* 2022, 144, 24, 10785–10797, <https://doi.org/10.1021/jacs.2c01768>.)

We acknowledge potential gaps in the initial description of our HTE system and have revised the manuscript to provide a comprehensive overview, distinguishing between automated and manual procedures. The revisions have been incorporated into the main text (*Page 8-9, Figure 2*) and the SI (*Figure S1, S5, and S6*). The selection of 42 samples per round was based on the microplate design, with the capacity for 48 samples (2ml vial) per plate, allowing for additional samples if necessary. Since our focus was on saturated solutions, intentional sample repetitions were not pursued. We recognized that precise volume transfer is critical for accurate solubility measurements and, to address this, ensured regular calibrations of our robotic system. Strict adherence to experimental protocols and the inclusion of control samples with known target concentrations were implemented to ensure reproducibility. Reliable measurements were confirmed when the observed concentration from qNMR closely matched the target concentration, typically within a relative standard deviation (RSD) of 5% (*Main text Page 10; Figure S3b*).

In our study, we underscore the fundamental role of solubility in determining overall cell performance, specifically within the context of NRFBs. Our research focuses on optimizing the solubility of redox active molecules in organic electrolytes, laying a strong foundation for understanding the interconnected properties that influence cell performance. Solubility, a pivotal factor in determining energy density, significantly impacts various key properties, including viscosity, conductivity, and chemical stability. Previous research has highlighted the hindrance caused by low solubility and poor stability of redox-active species in NRFBs (Mechanism-Based Development of a Low-Potential, Soluble, and Cyclable Multielectron Anolyte for Nonaqueous Redox Flow Batteries, *J. Am. Chem. Soc.* 2016, 138, 47, 15378–15384, <https://doi.org/10.1021/jacs.6b07638>; High-energy density nonaqueous all redox flow lithium battery enabled with a polymeric membrane, *Sci. Adv.* 2015;1:e150088, DOI: 10.1126/sciadv.1500886). While our manuscript primarily focuses on the implementation of a high-throughput experimentation platform combined with an active learning approach for the precise optimization of solubility, our results illustrate a direct link between enhanced solubility of redox active molecules and improved performance of NRFBs, as solubility stands as the key factor in designing redox-active materials for high-capacity RFBs (*Batteries* 2023, 9(10), 504; <https://doi.org/10.3390/batteries9100504>; *Materials Today Energy* 34 (2023) 101286; <https://doi.org/10.1016/j.mtener.2023.101286>).

In response to the comments about the use of electrochemical terms, we have made the necessary revisions to ensure accurate and consistent terminology throughout the manuscript. Additionally, we have enhanced the paper by incorporating more relevant references and ensuring that the English language usage meets the required standards for publication in Nature Communications.

Reviewer 2

Comment:

The manuscript by Noh et al. presents high-throughput screening of solubilities of 2,1,3-benzothiadiazole as a model ROM for organic redox flow batteries by a robotic platform, in which the authors adopt Bayesian optimization to assist the evaluation and exploration of unary and binary mixtures. The study demonstrates that AI-guided robotic platform can efficiently look for non-intuitive combinations of solvent electrolytes for energy applications. The work is of significance to the battery community, but the following concerns need to be addressed.

Response:

We appreciate reviewer's positive comments and summary. We resonate the same way with the reviewer.

Comment:

1. The authors should provide the brand and model of the automated deck and clearly state your own contributions, i.e. how you (1) interfaced the vision system and the qNMR with the deck and (2) developed the GPR model for checking the solubility of BTZ. These details would help highlight contributions to the field.

Response:

We thank the reviewer for pointing out the missing information. We have added the following information regarding the robotic platform in the Experimental Methods section (main text, Page 18): "The saturated solutions were prepared by a robotic platform (Big Kahuna, Unchained Labs) as shown in Fig. S1."

- (1) The integration of Keyence's CV-X vision system into our robotic platform was made seamlessly with the control software of the vision system being incorporated directly into the operation of our robotic platform, as depicted in Fig. S6a and S6b. To facilitate automated NMR sampling, we designed a 3D-printed NMR tube holder specifically tailored to fit the deck position, as shown in Fig. S6c and S6d.
- (2) Data obtained from NMR analysis are used for developing the GPR model for solubility prediction. Detailed description of our GPR model can be found in page 12 and 13 of the main text, and in Computational Methods (Main text, Page 21).

Fig. S6. Integration of (a) Keyence's CV-X vision system and (b) its control software in the operating system of our robotic platform. (c) Stand-alone and (d) on-deck picture of the 3D-printed NMR tube holder.

2. In Fig 3, it would be nice to include error bars to the obtained values.

Response:

We appreciate your feedback and understand your concerns regarding the reproducibility of our results and the lack of information regarding the number of repeated measurements and the uncertainty associated with each data point. We would like to provide further clarification on these issues. Through our previous work (<https://doi.org/10.1016/j.xcrp.2023.101633>), we determined that the main sources of uncertainty were related to the sample processing steps. Therefore, we focused on standardizing and optimizing these processes to minimize variability and increase the reliability of our results. As a result of the standardized sample processing, we found that the uncertainties associated with the saturated solutions became negligible (note: we assume that the uncertainty associated with the NMR instrument itself is negligible). Due to the identical nature of the saturated solutions, it was not necessary to repeat the samples for each data point. However, it is important to clarify that we included control samples in each round of real experiments when

we didn't repeat the samples. These control samples served as reference points to monitor the reliability of the sampling process and the analytical method used. By including control samples, we were able to track any potential variation that may have occurred during the experimental procedure. This approach provided us with an additional measure of confidence in the reliability and accuracy of our results.

To resolve this uncertainty question, we have considered your feedback and added more discussion to the revised main text (page 10): “To ensure reproducibility, we also employed two control samples (2 M and saturated BTZ solutions in ACN) in every batch, particularly when repeat testing was not possible. The consistency of solubility values in these control samples across multiple batches, with a relative standard deviation of less than 5%, as shown in Fig. S3b, validates the reliability and precision of our HTE approach, ensuring the generation of repeatable and high-fidelity data.”

3. How was the 8-hour sample stabilization time decided? Did you have a calibration curve or choose the value randomly? Since this step takes the longest time among all steps, optimizing it could potentially shorten the overall time, hence the screening rate.

Response:

Thank you for your insightful suggestion aimed at improving process efficiency. Regarding the decision on the 8-hour sample stabilization time, we refrained from conducting experiments for calibration curves due to our extensive list of over 2,000 binary solvents, each composed of diverse solvents with varying physicochemical interactions with our target solute, BTZ. To expedite the process toward dissolution equilibrium, we implemented vigorous agitation and employed a small sample size, guided by recommendations in equilibrium solubility measurement for ionizable drugs (ADMET and DMPK, 4(2), 117–178. <https://doi.org/10.5599/admet.4.2.292>).

Considering the relatively high solubilities (>0.1M) of redox-active molecules for RFBs, they inherently reach equilibrium much faster than poorly soluble molecules, as highlighted in the literature (Chem. Soc. Rev. 47, 69-103. 10.1039/c7cs00569e). Although equilibration typically takes a few hours, we opted for an 8-hour stabilization period to ensure thorough stabilization of all binary solutions. This decision was informed by our prior studies in aqueous systems (<https://doi.org/10.1016/j.xcrp.2023.101633>), quick stabilization time screening (Table S9, no obvious changes beyond 6 hours of stabilization), operational ease (one overnight per round of experiments), and insights from relevant literature (Alsenz J, Kansy M. High throughput solubility measurement in drug discovery and development. *Adv Drug Deliv Rev* 59, 546-567 (2007)).

While we acknowledge the validity of your suggestion to optimize the stabilization time based on solvent composition through calibration curves, this approach was not pursued in the current study. However, we recognize the potential for enhancing time efficiency and screening rates, and we appreciate this valuable recommendation for consideration in future research endeavors.

Table S9. Stabilization time comparison after mixing the solutions.

Solvent	Molarity (M, mol/L at 20 °C)		
	6 hours	8 hours	20 hours
DOX	5.50	5.47	5.48
PC	2.77	2.79	2.78
DMSO	4.30	4.26	4.31
CH	2.08	2.06	2.09
ACN	4.77	4.64	4.8

4. It is to be reminded that high solubility is not the only criterion for high energy redox flow batteries. Other physicochemical and electrochemical properties, such as the viscosity, stability, and conductivity, should be carefully evaluated after identifying the sample mixture.

Response:

Thank you for your insightful comments on the practical application of NRFB. We acknowledge the significance of various physicochemical and electrochemical properties, including viscosity, electrochemical, and thermal stability during operation, as critical factors for determining the efficiency of NRFB systems. While we recognize the importance of properties such as viscosity, stability, and conductivity, our primary focus on solubility is driven by its key role in determining the energy density of RFBs.

In response to your suggestion, we conducted additional experiments to measure the viscosity of pure solvents and saturated solutions, as illustrated in *Fig. S8*. Our findings revealed that viscosity was minimally affected by the concentration of BTZ. Therefore, solubility screening serves as the initial step in evaluating theoretical energy density. Subsequent screening stages will address electrochemical reaction rates (CV) and transportation properties (ionic conductivity and viscosity), among other factors, laying the foundation for future publications.

Considering your valuable feedback, we have incorporated the High-throughput Viscosity Measurement part into the Experimental Methods section (main text, Page 20): “We developed a high-throughput viscosity measurement workflow by integrating automated sampling on our robotics platform (100 μ L saturated solution into a 2 ml vial) with a high-throughput viscometer (VROC® initium one plus, RheoSense) (Fig. S8b). According to our analysis, viscosity displays minimal sensitivity to the concentration of BTZ, resulting in an increase of less than 2 cP in solutions. Notably, the majority of saturated solutions exhibits viscosity values below 2.5 cP, as illustrated in Fig. S8a.”

Fig. S8. (a) Steady shear viscosity as a function of solvent at 20 °C. (b) The high-throughput viscometer (VROC® initium one plus, RheoSense) used in this study.

5. By the way, the authors should correct all “physiochemical” to “physicochemical”, as the two words have different meanings.

Response:

We really appreciate reviewer’s catch. We have corrected it.

Reviewer 3

Comment:

Authors report on an integrated robotic ML guided setup to optimize solubility on a model redox additive. Solids handling is tricky in automated systems and authors have done a good job in closing the loop. A few comments below

Overall, the authors have done an impressive job to push on solids/liquids handling and closing the loop related to electrolyte optimization.

Response:

We appreciate reviewer’s nice comments and summary. We resonate the same way with the reviewer.

Comment

1. Figure 1 needs to be improved. The figure itself is not of the right quality and caption description does not tell me how the system is closed loop. Please make this a clear system flow diagram and provide detailed caption.

Response:

We appreciated on your concern about the quality of Fig.1, and we have updated the figure as followed to better illustrate our closed-loop workflow.

Fig 1. Schematic workflow of the closed-loop electrolyte screening process based on ML-guided high-throughput experimentation platform.

Comment

2. Figure 2f, recommend adding the acceleration factor as a second y-axis.

Response:

We thank the reviewer for the suggestion. We have updated Fig. 2g and added the 'Acceleration Factor' as the secondary axis.

Figure 2g Evaluated experimental time per sample for different solubility measurement methods. The data for automated 'excess solvent' method was estimated from the work of Shiri and co-workers (iScience 24, 102176, 2021).

Comments

3. I'd recommend dropping AI term used in the paper and modify it to ML or Active Learning as described in the abstract. There is very narrow set of machine learning done here.

Response:

Following the reviewer's suggestion, we have replaced most of AI terms with either AL (Active Learning) or ML (Machine Learning) where it is appropriate.

Comments

4. How do the authors plan to give the dataset for the measurements? I'd recommend having a clear schema in a csv file that can easily be ingested in standard programming frameworks. I'd recommend SMILES, SELFIES or CAS Number or some other unique representation for the molecules in the system.

Response:

We appreciate the reviewer's suggestions. We have provided the link to our datasets (in '.csv' and '.xlsx' formats) in the Data and Code Availability section. These datasets include SMILES representation of all investigated solvents and their corresponding measured solubilities.

Main text, Page 21:

Data and Code availability

The solubility database of BTZ in 218 single and binary solvents (including their SMILES representations), and the Python code used for Bayesian optimization can be found in the following repository:

<https://github.com/MolecularMaterials/COBOL/tree/main/cobol/CaseStudy/solubilityRedoxHTE>

Comments

5. Did you try other acquisition functions - Thompson sampling, upper confidence bound, in addition to EI?

Response:

We thank the reviewer for the suggestion. Based on the observation from our previous works (Doan HA, et al. Quantum Chemistry-Informed Active Learning to Accelerate the Design and Discovery of Sustainable Energy Storage Materials. Chemistry of Materials 32, 6338-6346 (2020); Agarwal G, Doan HA, Robertson LA, Zhang L, Assary RS. Discovery of Energy Storage Molecular Materials Using Quantum Chemistry-Guided Multiobjective Bayesian Optimization. Chemistry of Materials 33, 8133-8144 (2021)), expected improvement (EI) consistently outperforms other acquisition functions including Thompson sampling, upper confidence bound, and probability of improvement. We found the same to be true in this work, as the benchmark result on the known dataset of 98 solvents indicates that EI requires the least number of evaluations to identify the solvent with highest solubility (Fig. S4).

Fig. S4. Comparison of random selection versus Bayesian optimization using various acquisitions on the initial known dataset of 98 solvents. EI, UCB, POI, and TS stand for expected improvement, upper confidence bound, probability of improvement, and Thompson sampling.

We have also added Fig. S4 to the SI and the following sentence to the main text to justify the use of EI: (Page 14)

“Furthermore, different acquisition functions such as Thompson sampling, upper confidence bound, and probability of improvement can also be used with BO to identify the optimal solvent more quickly than random selection; however, EI is shown to have the best performance among them (Fig. S4).”

Comments

6. A few other non-technical comments: I'm a little disappointed that Library Studio Ver 9.2 (Unchained Lab) has been used for experiment design rather than some open-source framework. I recognize that the system was purchased, so this decision had to be taken but this makes it very difficult to reproduce the results.

Response:

We appreciate the reviewer's comment and recognize the significance of employing open-source frameworks to promote transparency and reproducibility. To address any challenges regarding result reproducibility, we have planned to upgrade our software. In collaboration with Unchained Labs, we are currently evaluating the transition from the older system to a new software version, which includes a user-friendly Python API (Python interface: "Drop in to easier and faster automation with Big Kahuna and Junior." <https://www.unchainedlabs.com/resource-library/?search=Drop+in+to+easier+and+faster+automation+with+Big+Kahuna+and+Junior>)

Comments

7. The title needs to be changed - please stress on the problem you are solving. The system is specifically focused on solubility - the title is overclaiming in terms of "electrolyte discovery". This system and other closed loop systems are only capable of optimization, discovery would require the ability to move through chemical space - which is beyond the capability of current ML-guided automated experimentation setups.

Response:

We appreciate the suggestion. We have updated the title to reflect our work more closely. The new title is:

"An Integrated High-throughput Robotic Platform and Active Learning Approach for Accelerated Discovery of Optimal Electrolyte Formulations"

Reviewer #4

Comments:

First, thank you to the authors for submitting their manuscript "An Integrated High-throughput Robotic Platform and Active Learning Approach to Accelerate Electrolytes Discovery" to the Nature Communications journal. The article presents an interesting integrated workflow combining high-throughput robotic experimentation and active learning to accelerate the discovery of organic electrolytes for redox flow batteries. The topic is relevant, and the approach is novel for accelerating the design of electrolytes for redox flow batteries.

Overall, this paper shows promise and requires revisions to address the comments above. If these issues are discussed thoroughly, the article could become suitable for publication after another round of review.

Response:

We appreciate reviewer's nice comments and summary. We resonate the same way with the reviewer.

Comment

1. Abstract: "Physiochemical property ..." -- "... Physicochemical property ..." & "... to generate and optimize..."  "... to optimise ...".

Response

We really appreciate reviewer's catch. We have corrected them.

Comments

2. Introduction:

(1) Shall we introduce the introduction with an “Introduction” section title? Naïve question regarding the expected manuscript’s format in the journal.

Response

We appreciate the reviewer’s suggestion. We have added “introduction” as a session title.

Comments

(2) “... cycle life ...”  “... lifecycle ...”, “... known as ROMs, ...”  “... (ROMs) ...”, “... high throughput ...”  “... high-throughput ...”, “... facile ...”  “... accessible ...”, “... The majority of high-throughput experimentation (HTE) ...”  Acronym already introduced earlier. Please use it. Overall, a third or fourth reading of the main text would have avoided many straightforward corrections. “... mixtures for a given organic molecule is practically infinite ...” -> “... mixtures for a given organic molecule is increased ...”

Response:

We appreciate the reviewer’s suggestion. We carefully reviewed the manuscript and addressed these problems based on your input.

Comments

(3) “... which increases the chances of finding a suitable combination but also makes the screening process more time-consuming and expensive even with HTE systems. ...” “... which makes the screening process more time-consuming and expensive even with HTE systems. ...”. The claim on an increased probability of positive findings with expanding the space of mixture possibilities isn’t supported by any data or principle here.

Response:

We appreciate the reviewer's insightful question. The concept of a 'synergistic effect' leading to enhanced solubility in certain binary solvent systems, compared to their intrinsic solubility in a single solvent, has been reported by Qiu J et al. as demonstrated in **Figure 21** (Qiu J, Albrecht J, Janey J. *Synergistic Solvation Effects: Enhanced Compound Solubility Using Binary Solvent Mixtures. Organic Process Research & Development* 23, 1343-1351 (2019)). This phenomenon is also recognized as a common strategy to improve electrolyte performance in lithium-ion batteries (Chem Rev 104, 4303-4417 (2004)). In our electrolyte discovery, we strategically explore specific binary solvent combinations to overcome limitations imposed by intrinsic solubility.

Exploring diverse solvent possibilities increases the probability of finding suitable combinations. However, we acknowledge that this expansion amplifies both the time and cost implications of the screening process. Even with a high-throughput experimentation (HTE) system, screening over 2000 samples remains time-intensive and costly. Therefore, it is imperative to develop an ML-guided HTE system for precise and efficient solubility data generation for ROMs in organic solvent systems.

Figure 21 Solubility profile of binary solvent mixtures for a solute. (Qiu J, Albrecht J, Janey J. Synergistic Solvation Effects: Enhanced Compound Solubility Using Binary Solvent Mixtures. *Organic Process Research & Development* 23, 1343-1351 (2019))

We have considered your feedback and added more discussion to the revised main text (page 6):

“Second, organic solvents can be utilized either in their pure form or as mixtures, offering nearly unlimited combinations. Indeed, solvent mixtures (e.g., binary solvents) are frequently used to enhance solubility and modify other properties through a “synergistic effect”.^{31, 32, 33} In such cases, the solute demonstrates higher solubility in a binary solvent compared to pure solvents.^{31, 32, 33} However, the large diversity of potential solvent mixtures also renders the screening process more time-consuming and expensive, even with HTE systems.^{33, 34} A strategic approach would be to develop an ML-guided HTE system for targeted and efficient solubility data generation for ROMs in organic solvent systems. Active learning (AL), and Bayesian optimization (BO) in particular, has been shown to be a reliable approach to accelerate the search for the desired electrolytes for energy storage applications.³⁵ Therefore, closed-loop experimental workflow guided by BO could be used to minimize HTE execution.^{36, 37, 38, 39}”

Comments

- In step 4 of the step-by-step workflow description, please briefly add the role of the acquisition function here as you introduce the surrogate model in step 3.

Response:

We appreciate the reviewer’s suggestion to clarify the description of our workflow. We have revised the description of the workflow and added the role of acquisition function in main text (Page 7) as follows:

“Subsequently, we apply an acquisition function within the BO framework to guide the selection of new samples, directing our evaluation based on the balance of predicted solubility values and associated uncertainties, i.e., fitness score, thereby streamlining the discovery and analysis of potential solvents.”

Comments

(5) "As the surrogate model...": "e.g." should be "i.e."

Response:

Thanks for the suggestion. The word has been replaced.

Comments

(6) As a general remark on references. Why don't you refer to “S. Matsuda, G. Lambard, K. Sodeyama, “Data-driven automated robotic experiments accelerate the discovery of multi-component electrolyte for rechargeable Li–O₂ batteries”, Cell Reports Physical Science, Volume 3, Issue 4, 2022, 100832, <https://doi.org/10.1016/j.xcrp.2022.100832>; for which the setup shares high similarities with the one detailed in the present manuscript.

Response:

The paper could be a good reference to show the efficient approach to discover the optimized electrolyte formulation using HTE and BO process. The paper has been cited as reference 35 in the main text (page 6):

“Active learning (AL), and Bayesian optimization (BO) in particular, has been shown to be a reliable approach to accelerate the search for the desired electrolytes for energy storage applications.³⁵”

Comments

(7) The authors claim the methodology determines "thermodynamic" solubility, but the details suggest it is closer to kinetic solubility. Thermodynamic solubility requires proper equilibrium, which is difficult to confirm in high-throughput experiments with short timescales. This distinction should be clarified.

Response:

Thanks for pointing out the potential confusion for readers. Our HTE approach, resembling the classic shake-flask method for measuring thermodynamic solubility, concurrently runs multiple samples to achieve high throughput. Although the equilibration time in the classic shake-flask method is fixed for true saturated solutions, we utilized the HTP methodology and robotic systems to process numerous parallel experiments for increased throughput. In our study, we prepared over 40 solutions in a single round through a rigorous dissolution process, allowing the saturated solutions (solid + liquid) to sit at 20°C for an equilibrium period of at least 8 hours.

Considering the relatively high solubilities (>0.1M) of redox-active molecules for RFBs, they inherently reach equilibrium much faster than poorly soluble molecules, as highlighted in the literature (Chem. Soc. Rev. 47, 69-103. 10.1039/c7cs00569e). Although equilibration typically takes a few hours, we opted for an 8-hour stabilization period to ensure thorough stabilization of all binary solutions. This decision was informed by our prior studies in aqueous systems (<https://doi.org/10.1016/j.xcrp.2023.101633>), quick stabilization time screening (Table S9, no obvious changes beyond 6 hours of stabilization), operational ease (one overnight per round of experiments), and insights from relevant literature (Alsenz J, Kansy M. High throughput solubility measurement in drug discovery and development. Adv Drug Deliv Rev 59, 546-567 (2007)). It is worth noting that, for some poorly soluble compounds, longer equilibrium times may be required.

Table S9. Stabilization time comparison after mixing the solutions.

Solvent	Molarity (M, mol/L at 20 °C)		
	6 hours	8 hours	20 hours
DOX	5.50	5.47	5.48
PC	2.77	2.79	2.78
DMSO	4.30	4.26	4.31
CH	2.08	2.06	2.09
ACN	4.77	4.64	4.8

Comments

(8) Solubility depends strongly on temperature, as the authors pointed out, but limited temperature data is provided. How was temperature controlled and measured? The impact of slight temperature variations should be considered.

Response:

We appreciate the reviewer's question and suggestion. Given our focus on identifying solvent formulations for enhancing the dissolution of redox active compounds, we maintained a fixed temperature at 20°C throughout our experiments. It is noteworthy that our HTE robotics platform is equipped with functional decks featuring a heating/cooling range from -20°C to 180°C, utilizing a high precision chiller system with a temperature accuracy of ± 0.01 °C. (*Huber, Petite Fleur model with dynamic temperature control system*). If required, our system can readily accommodate the study of temperature-dependent solubility.

Comments

3. Results and discussion:

(1) Figure 2-f needs more consistency in uncertainty quantifications. Because of this, it's hard to assess the viability of the presented methodology compared to the two others, even though the absolute difference with a manual experiment is still expected to be significant. Please upgrade the figure accordingly as far as possible. If the authors struggle to assert the uncertainty of a given methodology, the reason should be pointed out in the main text.

Response:

We appreciate your valuable feedback and understand the concerns you raised regarding the consistency in uncertainty quantifications. The purpose of Figure 2f (now Figure 2g in the revised manuscript) is to evaluate the experimental time per sample for different solubility measurement methods. In response to your comments, we have taken steps to enhance clarity by updating the figure and providing additional information in both the main text (page 9) and the SI (Table S1):

“With our automated HTE workflow, the total experimental time to finish the solubility measurement for 42 samples is ca. 27 hours (~39 minutes/sample, less time per sample with running more samples). As shown in Fig. 2g, this is more than 13 times faster than processing samples one by one manually using the ‘excess solute’ approach, which requires approximately 525 minutes per sample (see Table S1 for detail). While the screening speed of our HTE workflow based on the ‘excess solute’ method is comparable to that of the automated platform proposed by Shiri et al. (20 - 80 minutes/sample),²⁷ there are two important distinctions. First, we measured thermodynamic solubility whereas Shiri and co-workers used the ‘excess solvent’ method for kinetic solubility measurements. Second, our workflow processes 42 or more samples at once while Shiri et al.’s platform operates on one sample at a time.”

Figure 2g Evaluated experimental time per sample for different solubility measurement methods. The data for automated 'excess solvent' method was estimated from the work of Shiri and co-workers (iScience 24, 102176, 2021).

Table. S1. Summary of processing time for Manual Excess Solute and HTE Excess Solute (this work).

	Manual Excess Solute	HTE Excess Solute	Automated Excess Solvent
Programing	0	30	
Solution Preparation	10	330	
Stabilization	480	480	
Experimental runtime (min)			
NMR Sampling	15	120	
NMR Measurement	15	600	
Post-Analysis	5	60	
Total Process Time	525	1620	
Number of samples in a single batch	1	42	

Experimental runtime per sample (min/sample)	525	38.6	20-80*
---	-----	------	--------

*Shiri P, et al. iScience 24, 102176 (2021).

Comments

(2) “..., an the workflow ...”  “..., a workflow ...”

“(Fig. 2a) Powder and liquid dispensing systems (Fig. 2b and 2c) were integrated ...”  This is the main text. Please update the format you use to introduce the figures, e.g., “Powder and liquid dispensing systems (Fig. 2b and 2c) were integrated into the workflow (Fig. 2a) ...”

“... vison-tracking ...”  “... vision-tracking ...”

“... acetonitriltheACN) ...”?

Response

We really appreciate reviewer’s catch. We have corrected it.

- “... the NMR spectra were comparable to the target concentration ...” à Please be more specific in the comparison. Quantify it and share it in the main text.

Response

We appreciate the reviewer’s question and suggestion. We have included detailed description of our qNMR method in the Solubility Measurement via Quantitative H-NMR Spectroscopy section (Main text, Page 19-20):

“Prior to the production runs, we prepared two reference samples in acetonitrile (ACN) with target BTZ concentrations of 1.0 M and 2.0 M to evaluate the accuracy of our automated workflow and qNMR analytical method. The solubilities of BTZ calculated from NMR spectra were found to be 0.98 M and 1.98 M (Fig. S7), indicating the accuracy of our approach.”

-“... conditionstheur ...”?

Response

We really appreciate reviewer’s catch. We have corrected it.

- Fig. 3-a, b: Is it possible to assess the uncertainty on the measured solubility per solvent (3-a) and per batch (3-b)?

Response:

We appreciate your feedback and understand your concerns regarding the reproducibility of our results and the lack of information regarding the number of repeated measurements and the uncertainty associated with each data point. We would like to provide further clarification on these issues. Through our previous work (<https://doi.org/10.1016/j.xcrp.2023.101633>), we determined that the main sources of uncertainty were related to the sample processing steps. Therefore, we focused on standardizing and optimizing these processes to minimize variability and increase the reliability of our results. As a result of the standardized sample processing, we found that the uncertainties associated with the saturated solutions became negligible (note: we assume that the uncertainty associated with the NMR instrument itself is negligible). Due to the identical nature of the saturated solutions, it was not necessary to repeat the samples for each data point. However, it is important to clarify that we included control samples in each round of real experiments when we didn't repeat the samples. These control samples served as reference points to monitor the reliability of the sampling process and the analytical method used. By including control samples, we were able to track any potential variation that may have occurred during the experimental procedure. This approach provided us with an additional measure of confidence in the reliability and accuracy of our results.

To resolve this uncertainty question, we have considered your feedback and added more discussion to the revised main text (page 10): “To ensure reproducibility, we also employed two control samples (2 M and saturated BTZ solutions in ACN) in every batch, particularly when repeat testing was not possible. The consistency of solubility values in these control samples across multiple batches, with a relative standard deviation of less than 5%, as shown in Fig. S3b, validates the reliability and precision of our HTE approach in producing repeatable and high-fidelity data.”

- “... miscible. (Fig. S3). From ...”  “... miscible (Fig. S3). From ...”

Response

We really appreciate reviewer’s catch. We have corrected it.

Comment

- “... we considered a total of 11 relevant features derived from physicochemical properties and electronic structure calculations (DFT) of both the solvent and solute ...”  How did you choose the relevant features?

Response:

The choice of the 11 physicochemical and DFT-derived descriptors was inspired from previous work (Boobier et al. Nat. Commun. 2020) and selected by human experts. We have added the

following sentence in the main text (page 12) to clarify our choice of the descriptors used in this work:

“Since each solvent sample consists of BTZ and up to two solvents, we considered a total of 11 relevant features derived from physicochemical properties and electronic structure calculations (DFT) of both the solvent and solute (e.g., molecular weight and topological polar surface area of a solvent molecule, computed maximum and minimum partial charge of a solvated BTZ molecule) as features for the GPR model (see Table S4 for the complete list of features). The selection of such features was inspired by previous works^{42, 43} and further assessed by human experts.”

Comment

- “... the partition coefficient (logP) value of the solvent was identified as the most important feature of the GPR model ...”  Is this statistical correlation of the solubility with the LogP consistent with the actual physicochemical knowledge? And if yes, why? Can you interpret the result of this feat're's importance distribution? As well as for 'TPSA' and the 'HOMO' level.

Response:

We thank the reviewer for the valuable comment. The correlation between solubility and logP_{solv} is indeed consistent with our physicochemical knowledge of the relationship between polarity and solubility (Yalkowsky et al.). LogP_{solv} is the octanol-water coefficient of a solvent and can be used to account for the difference between the polarity of that solvent and water. If we consider the polarity of water as the reference, and since the difference between the polarity of BTZ and water is a constant, logP_{solv} should reflect the polarity difference between BTZ and any solvent system of interest. Similarly, the TPSA_{solv}, or the Topological Polar Surface Area of a solvent molecule, indirectly indicates the local polarity difference between BTZ and the solvent. We currently do not have an explanation for the correlation between HOMO energy of BTZ and its solubility.

We have added our reasonings for the correlation between solubility and the most important features derived by ML models as follows (page 12-13 of the main text):

“Here, logP_{solv} represents the octanol-water coefficient for a solvent, serving as a means to assess the polarity disparity between that solvent and water. If we consider the polarity of water as our baseline, given that the polarity variance between BTZ and water remains constant, then logP_{solv} effectively characterizes the polarity distinction between a solvent and BTZ. Similarly, TPSA_{solv}, denoting the Topological Polar Surface Area of a solvent molecule, indirectly offers insights into the localized polarity contrast between a solvent and BTZ. These findings are consistent with the current knowledge of solubility as a function of polarity, as in the general solubility equation proposed by Yalkowsky et al.^{42,43}”

Comment

- Can you explain the origin of the error bars in Fig. 4-a? Possibly as an intrinsic consequence of the GPR model?

Response:

We thank the reviewer for the suggestion. The error bars in Fig. 3a (*It is Fig. 4a in our original manuscript*) represent one standard deviations from the means, which are indeed calculated from the GPR model. We have added the following details to the caption of Fig. 3a (main text, Page 13) to explain the origin the of the error bars:

“Fig 3. (a) Parity plot of GPR-predicted molarity versus measured values from experiments. The error bars represent one standard deviation...”

Comment

- “... Bayesian optimization (BO) ...”  You don't need to re-introduce the acronym here as already done in the introduction.

Response:

We appreciated the correction. All of the repeated words were replaced to acronym, BO.

Comment

- “... average BO identifies DOX:DMF @ 0.6:0.4 after suggesting a total of 17 out of 98 solvents for solubility evaluation ...”  What's the uncertainty on 17 (i.e. from Fig. 5-b) as you indicated it for the random selection (50+/-27)? Consistent uncertainties quantification is the root of statistically well-supported comparisons.

Response:

We thank the reviewer for pointing this out. The uncertainty of the BO approach is ± 11 , which is the error bar in Fig. 4b. We have added this missing value to the main text (page 14) accordingly:

“Our results indicate that on average BO identifies DOX:DMF@0.6:0.4 after suggesting a total of 17 ± 11 out of 98 solvents ...”

We have also updated Fig. 4b (main text, Page 15) to indicate the uncertainty values corresponding to BO and random selection.

Fig. 4. (a) Schematic representation of Bayesian optimization workflow algorithm for accelerated screening of binary solvents with desired solubility of BTZ. (b) Comparison between Bayesian optimization and random selection for the number of required evaluated solvents before identifying the one with highest solubility of BTZ from the 98 solvent dataset. The height of the color bar and error bar represent the mean and standard deviation of 100 trials.

Comment

- The data analysis comparing random selection and Bayesian optimisation needs more rigour. The authors should perform statistical tests (e.g., compute p-value) to determine if the difference between both methods is significant. Error bars in Fig. 5-b are indeed substantial.

Response:

As the reviewer recommended, we have performed T-test to compute the p-value for the datasets obtained by Bayesian optimization and random selection. We find the p-value to be equal to 1.17×10^{-20} , indicating that the performance difference between Bayesian optimization and random selection is statistically significant. We have also added the following sentence to the main text (page 14) to further justify the comparison between BO and random selection:

“We also performed t-test on the two distributions and obtained a p-value of 1.17×10^{-20} , indicating that the performance improvement of BO over random selection is statistically significant.”

Comment

“... as shown Fig. 5a ...”  “... as shown in Fig. 5a ...”

Response:

We really appreciate reviewer’s catch. We have corrected it.

Comment

- Feature generation: “In addition, we carried out first-principles simulations of solvated BTZ molecules in different solvents to compute solute-related descriptors such as ...”  Can you assess the time needed to perform the DFT calculations for the initialisation set and per batch? The time should be added to the time needed for the automated protocol in the manuscript for the initialisation set and per batch. In this case, the computation time needed to apply the BO should be relatively insignificant.

Response:

We thank the reviewer for the opportunity to clarify this more. To avoid building complex solvation models consisting of various binary solvent mixtures, we also applied feature mixing to the DFT-generated descriptors. Namely, a descriptor for a binary mixture is the combination of the same descriptor calculated for each unary solvent weighted by their mol fractions. Since all the unary solvents are known, their corresponding DFT-derived descriptors can be computed ahead of time. Therefore, the time requirement to generate first-principles descriptors for binary solvents during the BO workflow is trivial.

Comment

- “... exploitation and exploration ...”  Please be a bit more specific on what those terms mean for non-specialists in the field of ML.

Response:

We have added the following details regarding exploitation and exploration in the description of the ML model in the Computational Methods section (main text, Page 22):

“In eq. 4 and 5, a constant ε value of 10^{-2} was used to balance the trade-off between exploitation (pursuing the trend of the current best estimates) and exploration (diversifying the search to avoid local optima)”

REVIEWERS' COMMENTS

Reviewer #2 (Remarks to the Author):

The authors have addressed my previous concerns. Therefore, I recommend it for publication.

Reviewer #3 (Remarks to the Author):

Authors have done a tremendous job of addressing the comments raised in the previous round of review. The new version is greatly improved.

Reviewer #4 (Remarks to the Author):

First, thank you to the authors for updating their manuscript "An Integrated High-throughput Robotic Platform and Active Learning Approach to Accelerate Electrolytes Discovery", according to the review.

Overall, the suggested corrections and amendments have all been properly considered, and the questions have all been extensively considered and answered. The article has gained clarity on the methodologies used for the sake of the reproducibility of the study, and the quantification of uncertainties has been properly managed, giving the article statistical support for the findings. The article may now become a reference to coming high-throughput robotic platform initiatives.

Thank you to the authors for their effort and time in this update.